# Precision Medicine through Next-Generation Sequencing in Inherited Eye Diseases in a Korean Cohort

**DOI:** 10.3390/genes13010027

**Published:** 2021-12-23

**Authors:** Dabin Moon, Hye Won Park, Dongheon Surl, Dongju Won, Seung-Tae Lee, Saeam Shin, Jong Rak Choi, Jinu Han

**Affiliations:** 1Department of Medicine, Yonsei University College of Medicine, Seoul 03722, Korea; moondabin200@yonsei.ac.kr; 2Department of Ophthalmology, Konyang University College of Medicine, Daejeon 35365, Korea; phwgj85@kyuh.ac.kr; 3Department of Ophthalmology, Institute of Vision Research, Severance Hospital, Yonsei University College of Medicine, Seoul 03722, Korea; dhsurl@naver.com; 4Department of Laboratory Medicine, Severance Hospital, Yonsei University College of Medicine, Seoul 03722, Korea; wdjbabo@yuhs.ac (D.W.); lee.st@yuhs.ac (S.-T.L.); saeam0304@yuhs.ac (S.S.); cjr0606@yuhs.ac (J.R.C.); 5Department of Ophthalmology, Institute of Vision Research, Gangnam Severance Hospital, Yonsei University College of Medicine, Seoul 06273, Korea

**Keywords:** precision medicine, inherited eye disease, next-generation sequencing, genetic testing

## Abstract

In this study, we investigated medically or surgically actionable genes in inherited eye disease, based on clinical phenotype and genomic data. This retrospective consecutive case series included 149 patients with inherited eye diseases, seen by a single pediatric ophthalmologist, who underwent genetic testing between 1 March 2017 and 28 February 2018. Variants were detected using a target enrichment panel of 429 genes and known deep intronic variants associated with inherited eye disease. Among 149 patients, 38 (25.5%) had a family history, and this cohort includes heterogeneous phenotype including anterior segment dysgenesis, congenital cataract, infantile nystagmus syndrome, optic atrophy, and retinal dystrophy. Overall, 90 patients (60.4%) received a definite molecular diagnosis. Overall, NGS-guided precision care was provided to 8 patients (5.4%). The precision care included cryotherapy to prevent retinal detachment in *COL2A1* Stickler syndrome, osteoporosis management in patients with *LRP5*-associated familial exudative vitreoretinopathy, and avoidance of unnecessary phlebotomy in hyperferritinemia-cataract syndrome. A revision of the initial clinical diagnosis was made in 22 patients (14.8%). Unexpected multi-gene deletions and dual diagnosis were noted in 4 patients (2.7%). We found that precision medical or surgical managements were provided for 8 of 149 patients (5.4%), and multiple locus variants were found in 2.7% of cases. These findings are important because individualized management of inherited eye diseases can be achieved through genetic testing.

## 1. Introduction

Clinical genetic testing with next-generation sequencing (NGS) has been gaining popularity of late [1,2]. Many inherited eye diseases, such as congenital cataracts or retinitis pigmentosa, exhibit considerable genetic heterogeneity; therefore, the use of NGS technology enables a larger number of patients to obtain a molecular diagnosis for their disease. Accurate molecular diagnosis is important for acquiring information regarding visual prognosis and guiding future family planning. However, genetic diagnosis generally has not led to a change in treatment or disease management. Since the US Food and Drug Administration approved gene therapy for Leber’s congenital amaurosis with biallelic *RPE65* variants in December 2017, genetic testing has gained greater attention. In addition, clinical trials are currently underway on gene therapy for multiple inherited retinal diseases, such as achromatopsia or choroideremia, which were generally regarded as incurable in the past [3]. 

The American College of Medical Genetics and Genomics has recommended reporting of incidental findings in clinical exome sequencing for a total of 59 genes [4]. Most of these disorders are rare life-threatening conditions, and *RB1* (retinoblastoma) was the sole vision-related gene. However, clinicians also should focus on genetic causes of less threatening, but treatable disorders, which require specific treatment [5]. These include Little’s syndrome, a genetic form of pseudo-aldosteronism, which can be treated with triamterene. Accurate diagnosis of syndromic ciliopathy through exome sequencing leads to early detection of renal disease before acute presentation [6]. In addition, dietary modification or preventive therapy can alter disease progression in patients with syndromic congenital cataracts [7]. As indicated by these examples, there is great potential for genetic testing to enhance patient care [8,9]. However, little is known regarding medically or surgically actionable genes in the field of ophthalmology.

Genetic testing is expensive, indeterminate in a large number of cases, and often not covered by health insurance. It remains controversial whether routine massive parallel sequencing in inherited eye diseases is warranted or not when clinical phenotype matches a single genetic cause [10]. Previous studies demonstrated the diagnostic accuracy of NGS in inherited eye diseases, but their usefulness regarding treatment has been not discussed much [11,12,13]. Here, we report the use of targeted-panel NGS to reach a clinical diagnosis and investigate whether it altered treatment in patients with inherited eye diseases.

## 2. Materials and Methods

### Recruitment and Selection of Patients with Inherited Eye Diseases

The cohort for this study included 149 consecutive unrelated patients with inherited eye diseases, with or without systemic conditions, who agreed to undergo genetic testing between 1 March 2017 and 28 February 2019. All patients underwent ophthalmologic examinations, which included slit-lamp examination, determination of the presence and type of nystagmus, determination of the presence of other systemic symptoms, fundus examination, and measurement of visual acuity. If indicated, electroretinography (ERG, Reti-port, Roland Consult, or RetEval, LKC Technologies, Gaithersburg, MD, USA) was performed in accordance with the standards of the International Society for Clinical Electrophysiology of Vision. ERG used skin electrodes in children younger than 6 years; sedation with oral chloral hydrate was used in patients younger than 2 years. Spectral-domain optical coherence tomography (OCT, Heidelberg Engineering, Heidelberg, Germany) was performed when applicable. Informed written consent was provided by patients and/or parents, and peripheral blood samples were obtained for genetic analysis from all patients. This study was approved by Severance Hospital Institutional Review Board (4-2018-0436) and adhered to the tenets of the Declaration of Helsinki.

For the customized NGS panel, we selected 429 genes known to cause inherited eye diseases, based on our assessment of literature reviews, the RetNet database (https://sph.uth.edu/Retnet/, accessed on 10 January 2017), and the Online Mendelian Inheritance in Man database (https://www.ncbi.nlm.nih.gov/omim) (Appendix A); we also included deep intronic or regulatory regions known to cause inherited eye diseases (Appendix A). Target enrichment was performed with custom-designed RNA oligonucleotide probes and target enrichment kit (Celemics, Seoul, South Korea). Sequencing and bioinformatics analyses were performed as described previously [12]. Briefly, pooled libraries were sequenced using a NextSeq 550 sequencer (Illumina, San Diego, CA, USA) and the NextSeq Reagent Kit, version 2 (300 cycles). For each sample, quality metrics were calculated using the FastQC software and TEQC package. Sequences were aligned to the hg19 reference genome using BWA-aln. [14] Single nucleotide variants and small insertions or deletions were called and crosschecked using GATK version 3.8.0 with Haplotypecaller [15] and VarScan version 2.4.0. Each variant suspected to be pathogenic, likely pathogenic, or variant of uncertain significance (VUS) was confirmed by visual inspection of the bam file using the Integrated Genomics Viewer 2.3 software. Split-read based detection of large structural variations was conducted using Pindel [16] and Manta [17]. Read-depth based detection of copy number variation (CNV) was conducted using ExomeDepth version 1.1.10. [18], followed by visualization using a base-level read depth normalization algorithm designed by the authors. CopywriteR version 2.9.0 was used with a 1 Mb window option for off-target analysis and chromosomal CNV detection [19].

The pathogenicity of missense variants was predicted using 5 in silico prediction algorithms, including SIFT, PolyPhen2, FATHMM, and CADD [20]. Splice site analysis was performed using the MaxEntScan, NNSPLICE (Neural Network Splice Prediction), Human Splice Finder, GeneSplicer, and SpliceFinder-like algorithms implemented in the Alamut Visual software (Interactive Biosoftware, Rouen, France) and deep-learning-based SpliceAI [21,22]. The variant was filtered by genome aggregation database (gnomAD) and Korean Reference Genome database. The interpretation of variants followed the 5-tier classification system recommended by the American College of Medical Genetics and Genomics and the Association for Molecular Pathology [23]. For the purpose of this study, patients were grouped in the following 2 categories: (1) definite molecular diagnosis: patients with pathogenic or likely pathogenic disease-associated variant(s), for whom phenotypes were exactly matched to the genotypes; (2) unsolved: all other patients for whom no pathogenic or likely pathogenic variants existed.

For small nucleotide variations, pathogenic and likely pathogenic variants, as well as variants of uncertain significance (VUS), that required parental study were further examined by Sanger sequencing on a 3730 DNA Analyzer with the BigDye Terminator v3.1 Cycle Sequencing Kit (Applied Biosystems, Foster City, CA, USA). Sequencing data were aligned against appropriate reference sequences and analyzed using the Sequencher 5.3 software (Gene Codes Corp., Ann Arbor, MI, USA). Large exonic deletions and duplications were confirmed by using a multiplex ligation-dependent probe amplification kit (MRC Holland, The Netherlands) whenever necessary. Chromosomal CNVs detected by the off-target analysis were validated by Affymetrix Cytoscan 750K array (Genome build, hg19). Affymetrix Cytoscan 750K array contains 750,000 markers for copy number analysis, including 200,000 SNPs and 550,000 polymorphic probes. Data analysis was performed using Chromosomal Analysis Suite software version 1.2.2.

## 3. Results

### 3.1. Patient Demographics

The systemic and ophthalmic features of the 149 unrelated are listed in online Appendix A. Among 149 patients, 99 (66.4%) were male, and 38 (25.5%) had family histories of similar phenotypes (Appendix A). All patients were of a single ethnicity (Korean) except 1 Vietnamese; none of the patients were of consanguineous parentage. The average age at the time of genetic testing was 14.3 ± 16.5 years (Appendix A). Systemic phenotypes, such as developmental delay, cerebellar atrophy, cleft palate, or facial dysmorphism were noted in 26 patients (17.5%). Infantile nystagmus was presented in 103 patients (69.1%). The cohort was phenotypically heterogeneous. Six were congenital cataracts, 10 patients were anterior-segment dysgenesis including *PAX6*-related phenotype, 48 were retinal dystrophy, and 85 were neuro-ophthalmic condition such as optic atrophy, albinism, or *FRMD7*-related infantile nystagmus (Figure 1B).

### 3.2. Molecular Diagnostic Rate and Personalized Medicine through Targeted Next-Generation Sequencing

Targeted NGS identified definite diagnostic variants in 90 patients (60.4%, 95% CI, 52.4–67.9%, Figure 1A and Table 1). The other 59 patients (39.6%) remained unsolved after targeted NGS. Among solved 90 patients, 25 patients were previously reported by our group [12,24,25,26,27,28,29,30]. The molecular diagnostic rate was higher in patients with family history than in singleton cases (86.8% vs. 51.4%, *p* < 0.001). The molecular diagnostic rate was also higher in patients with systemic phenotype than patients with a non-syndromic phenotype (73.9% vs. 57.9%, *p* < 0.001). The molecular diagnostic rates varied among disease groups. Sixty-six percent were solved in congenital cataracts, 90% (9/10) were solved in anterior segment dysgenesis, the causative variants were found in 75% (36/48) in the retinal dystrophy group, and 48.2% (41/85) were solved in the neuro-ophthalmic group (*p* < 0.004).

We identified a total of 118 disease-associated variants in 90 patients. *FRMD7*, *PAX6*, and *GPR143* variants appeared to be frequently mutated in our cohort. Among 118 variants, 47 (39.8%) were novel variants. A total of 7 CNVs were detected, and 1 non-coding pathogenic variant was found. Overall, medically or surgically actionable genes were found in 8 patients (5.4%), and they received NGS-guided precision care after molecular diagnosis (Table 2 and Figure 1). The initial clinical diagnosis was revised in 22 patients (14.8%). Among 149 patients, unexpected chromosomal large CNVs (n = 2) and multi-locus variations (n = 2) were noted in 4 patients (2.7%) (Figure 2 and Appendix A).

### 3.3. Case Examples

Case study 1 is important with respect to patient-specific treatment. The mother of the proband (P12) had a history of bilateral retinal detachment and had undergone vitrectomy by her twenties. The proband had exhibited high myopia since a young age, and she had micrognathia and a cleft palate. Dilated fundus examination showed bilateral thick vitreous strands at the retinal periphery (Figure 3). Targeted-panel NGS identified a *COL2A1* c.2680-3C>G non-canonical splice site variant. The variant was not predicted by CADD to be deleterious (CADD score: 8.353), but it was predicted by Human Splice Funder 3.1 to cause a new splice site acceptor (wild type 63.92 vs. mutant 92.86: +45.28% variation); this was supported by the results from 2 other splice site prediction programs and SpliceAI (Appendix A). Prophylactic cryotherapy was applied to prevent future risk of retinal detachment.

Case study 2 is an example of avoiding unnecessary treatment by using targeted NGS testing. The proband (P38) had congenital cataracts in early childhood. The mother and grandmother also had histories of congenital cataracts and had undergone bilateral cataract surgeries during their twenties. His mother had elevated ferritin levels, and repeated phlebotomies had been performed to reduce the burden of iron overload. Targeted-panel NGS revealed c.-168G>T in the *FTL* gene (Figure 4). This upstream regulatory variant was previously reported as pathogenic in hyperferritinemia-cataract syndrome [31], therefore, we included this region in the design of the targeted panel (Appendix A). Individuals with hyperferritinemia-cataract syndrome do not have an excess of iron; repeated phlebotomies in these patients may cause anemia. Correct diagnosis of hyperferritinemia-cataract syndrome is important to avoid unnecessary treatments or invasive procedures, such as repeated phlebotomies or liver biopsies.

Case study 3 illustrates that chromosomal CNV was identified after targeted NGS. A 4-year-old girl (P86) presented at our clinic for strabismus. She had micrognathia, developmental delay, and hirsutism (Figure 5A). Dilated fundus examination showed optic atrophy in both eyes (Figure 5B). At the age of 8 years, best-corrected visual acuity was 20/50 in both eyes. Targeted NGS showed 12p12 microdeletion (Figure 5C), and this chromosomal CNV was confirmed by microarray. This region included the *SOX5* gene, and it was known to cause non-progressive optic atrophy and strabismus [32]. Regular monitoring of cardiac function is recommended because *ABCC9* gene deletion (Appendix A) is known to cause pericardial effusion and cardiomegaly [33].

Case study 4 is an example of dual diagnosis in a single patient. A 4-year-old male came to our hospital for the evaluation of upbeat nystagmus. His best-corrected visual acuity was 20/150 in the right eye and 20/100 in the left eye, and mild myopia was noted. Targeted next-generation sequencing showed a novel heterozygous c.1910+1G>A variant in the *CACNA1F* gene and a novel heterozygousc.3833G>A:p.(Trp1278*) variant in the *LRP5* gene. Detailed phenotyping was consistent with the phenotype, and a dual diagnosis was made (Figure 6).

## 4. Discussion

Molecular genetic tests based on NGS technology are widely used these days. However, patient-specific treatment is not available in most inherited genetic diseases. Our study revealed that the diagnostic yield of targeted NGS in inherited eye diseases was approximately 60%. Among patients who underwent genetic testing, about 5% of patients received precision care through genetic testing, and about 2.7% of patients had diagnoses that involved two or more diseases loci. Our result was consistent with a previous study, which reported that multiple molecular diagnoses could be made in 4.9% of cases in exome sequencing [34]. It is likely that many ophthalmologists do not have knowledge regarding which genetic profiles are medically or surgically actionable, and some may believe that genetic testing does not lead to a change in treatment. Our study showed that accurate molecular diagnosis has an impact on the understanding of molecular mechanisms, and it will help us to provide genotype-driven tailored investigation, prevent secondary complications or associated medical conditions, and avoid unnecessary treatment.

It has been reported that the molecular diagnostic rate varied among different disease groups [35]. Generally, the molecular pickup rate of NGS is higher in inherited eye diseases than other pediatric neurological diseases. In our study, anterior segment dysgenesis had the highest diagnostic rate. This high diagnostic rate is inconsistent with a previous study [36]. This could be related that most of the anterior segment dysgenesis were *PAX6*-related phenotypes except for one patient with Peter’s anomaly. In congenital cataracts, we identified causative variants in two-thirds of patients. This finding was consistent with a previous study [1]. In unsolved patients, one patient (P96) with oculocutaneous albinism had a heterozygous c.1025C>T:p.(Ala342Val) variant in the *SLC24A5* gene. Because the discovery of regulatory or deep intronic variants receives more attention [37], further studies with genome sequencing are needed to find non-coding variants in this gene.

Some patients with rare inherited eye diseases may receive an incorrect diagnosis or require many years to reach a definitive diagnosis [11,38,39]. Young children are typically not sufficiently cooperative to complete eye examinations or other diagnostic tests [12]. Children with inherited eye diseases may harbor serious medical illnesses associated with ocular phenotypes. Occasionally, they may undergo unnecessary brain imaging or numerous investigations before a correct diagnosis [12]. For example, FEVR is known to be caused by six different genes: *FZD4*, *TSPAN12*, *LRP5*, *NDP*, *ZNF408*, or *KIF11* [40]. Defects in one of these genes cause impaired development of the retinal vasculature by impacts on Norrin signaling pathways. Among these six genes, only the *LRP5* variant causes osteoporosis by impacting Wnt signaling pathways [41]. Bone densitometry and calcium supplements are helpful in patients with *LRP5*-associated FEVR.

Previous studies reported that molecular diagnostic rates of inherited retinal diseases varied between 50% and 76% [2,11,13]. Our results were consistent with these results. Stone et al. demonstrated that a tier-based approach in genetic testing increased the molecular diagnostic rate [13]. A tier-based approach can detect variants in *RPGR* exon 15, which is poorly covered by exome or genome sequencing. We agree that a tier-based approach is more cost-effective and has greater diagnostic yields, but it requires a high degree of clinical expertise [42]. Moreover, Posey et al. demonstrated that about 5% of patients with inherited genetic diseases had multi-locus genomic variations [34]. Even when the clinical context indicates a single genetic etiology, targeted NGS can be considered first tier because patients may harbor other ophthalmic conditions, which are not recognized by clinical examination. Consugar et al. also reported that panel-based NGS is more sensitive than exome sequencing for variant detection in known genes [43]. Targeted-panel NGS can also detect known deep intronic or regulatory variants, as shown in our case (P38). Initial analysis of genes well known to be associated with a particular phenotype will improve the positive predictive value and reduce the likelihood of false ascertainment [44]. However, it is necessary to update the target panel regularly. 

Some inherited eye diseases can be treatable and actionable (Table 3). For example, nystagmus associated with episodic ataxia type 2 caused by *CACNA1A* pathogenic variant can be ameliorated with acetazolamide, and Leber congenital amaurosis caused by bi-allelic *RPE65* variants can be treated with gene therapy [45,46]. Retinal detachment in Stickler syndrome type 1, which is caused by *COL2A1* pathogenic variants, can be prevented by prophylactic cryotherapy [47,48]. Prophylactic cryotherapy is not indicated in Stickler syndrome type II and III. High myopia and vitreous abnormalities are not specific to Stickler syndrome, and similar phenotypes are also noted in Knobloch syndrome [49]. The phenotypic complexity of rare genetic diseases may present a challenge to the clinician. Some inherited eye diseases are very rare and even experienced ophthalmologists who specialize in genetic eye diseases may not have seen them before. The suggested medically or surgically actionable genes in ophthalmology (Table 3) will help ophthalmologists or laboratory physicians to gain awareness of actionable genes in the field of ophthalmology.

We could not determine the pathogenicity of variants in several cases. For example, we found a *CASK* c.2147A>C:p.(His716Pro) variant in a patient (P91) with nystagmus and no intellectual disability. This variant was predicted to be deleterious by in silico prediction (CADD:27.6, PoplyPhen2: 0.983, SIFT: 0.01). A null variant in *CASK* (calcium/calmodulin-dependent serine protein) induced variable intellectual disability, with or without nystagmus [50]. CASK promotes FRMD7 co-localization at the plasma membrane, where it enhances CASK-induced neurite length [51]. Although the *CASK* c.2147A>C variant was absent in a population database, a functional analysis should be conducted to confirm the pathogenicity of this missense variant. Because of efforts to construct large population datasets, such as the gnomAD, many variants that had been considered pathogenic are now reclassified as benign or likely benign [52]. For precision medicine to succeed, it also needs to be more accurate [9]. Comprehensive phenotyping, accurate bioinformatics analysis including CNV detection, and cautious interpretations are essential parts of genetic diagnosis. Physicians should also know the limitations of NGS that it cannot reliably detect variants in high GC-rich regions, segmental duplication, or short tandem repeats. We agreed that variants should be considered uncertain until proven otherwise [44].

Our study has several limitations. First, the design of the study was retrospective, but our cases were collected consecutively by a single experienced pediatric ophthalmologist. Second, the majority of our patients were of a single ethnicity (Korean). Third, although our panel included 429 genes and known deep intronic variants associated with inherited eye diseases, including congenital cataracts, new recently discovered genes may have been missed in this panel-based sequencing approach. We assessed medically or surgically actionable genes in the field of ophthalmology based on literature searches, Genetic Home References, and the authors’ clinical experiences. More thorough investigations or reviews of systematically curated databases are needed.

## 5. Conclusions

In conclusion, our study demonstrated that NGS can help make an accurate diagnosis, provide individual-specific treatment, and perform targeted medical investigations. Our approach helps to avoid numerous medical investigations or unnecessary treatments. Meticulous eye examinations are not easy for young patients, and some patients may harbor multi-locus genetic variations in two unrelated genes. Therefore, targeted NGS can be considered a first-line diagnostic tool in inherited eye diseases, even if a single genetic cause is suspected. With an understanding of genetic mechanisms in inherited eye diseases, it would be easier to provide more specific treatments. Until now, most inherited eye diseases lack interventions that can prevent or modify the progressive course. In the near future, more patients will receive precision care if new gene therapy or pharmacogenetics studies are undertaken and proven to be effective.

## Figures and Tables

**Figure 1 genes-13-00027-f001:**
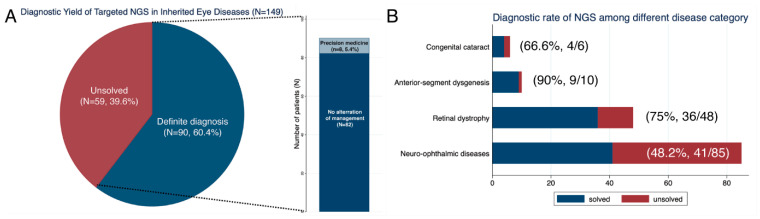
Diagnostic rate of targeted next-generation sequencing (NGS) in inherited eye diseases. (**A**) Overall diagnostic rate of targeted NGS and the proportion of patients who received precision medicine after genetic testing. (**B**) Diagnostic rate of targeted NGS among each disease category.

**Figure 2 genes-13-00027-f002:**
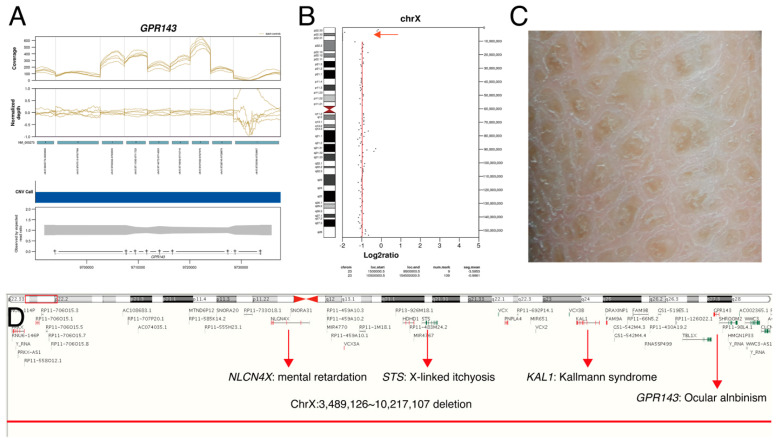
Copy number variations from targeted next-generation sequencing in a patient with ocular albinism (P47). A 5-year-old male had nystagmus since early infant. His past medical history was significant for congenital hypothyroidism, developmental delay, intellectual disability, and ichthyosis. Dilated fundus examination showed depigmented fundi and multichannel visual evoked potential showed chiasmal misrouting: (**A**) Normalized depth analysis on *GPR143* region showed a whole gene deletion of *GPR143*; (**B**) chromosomal copy number variations called by off-target analysis using CopywriteR version 2.9.0 showed absence of read depth on Xp22.3 (red arrow); (**C**) dry, thickened, scaly skin was noted; (**D**) ChrX:3,489,126_10,217,107 deletion was confirmed by array comparative genomic hybridization.

**Figure 3 genes-13-00027-f003:**
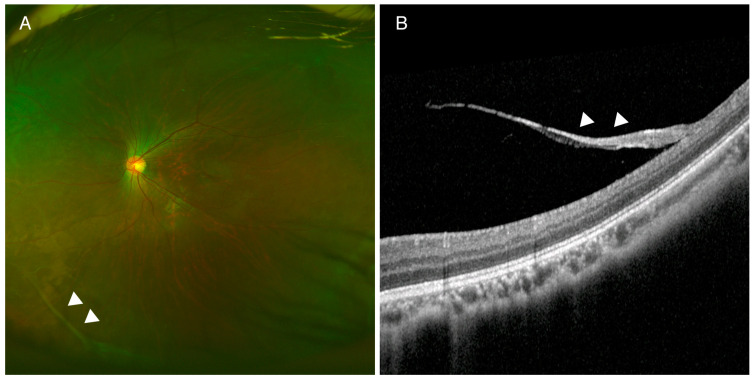
Fundus photography and optical coherence tomography (OCT) in a patient with Stickler syndrome (P12). (**A**) Peripheral vitreous degeneration was noted in wide fundus photography. (**B**) Spectralis OCT showed a thick vitreous band (arrowhead) attached to the retina. Targeted next-generation sequencing revealed a *COL2A1* c.2680-3C>G variant, predicted to cause a new splicing acceptor site in intron 40. Prophylactic cryotherapy was applied to prevent retinal detachment.

**Figure 4 genes-13-00027-f004:**
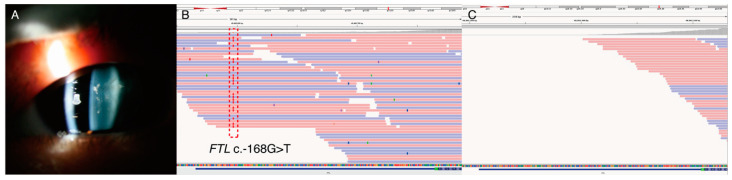
(P38) A 7-year-old male had congenital cataracts in both eyes. Laboratory examination showed elevated ferritin level 1205.3 ng/mL (reference range: 23.9~336.2 ng/mL) without iron overload. The inheritance pattern was consistent with autosomal dominant. (**A**) Slit-lamp examination showed congenital cataracts since early childhood. (**B**) Integrative genomic viewer showed *FTL* c.-168G>T variant in upstream to starting codon. This region was included in our panel, and the variant was previously reported as pathogenic. (**C**) The 5′ untranslated region of exon 1 in *FTL* gene was not covered in commercially available exome sequencing. Molecular diagnosis of hyperferritinemia-cataract syndrome enables us to avoid unnecessary investigations or treatment such as repeated phlebotomy or liver biopsy.

**Figure 5 genes-13-00027-f005:**
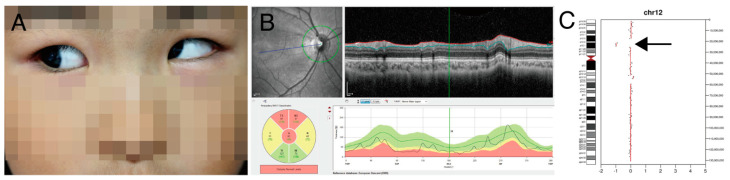
(P90) Chromosomal copy number variations analysis using off-target reads led to precision medicine (**A**) 8-year-old female showed strabismus and hirsutism was noted in philtrum area. (**B**) Spectral-domain optical coherence tomography showed optic nerve atrophy in the right eye. (**C**) Chromosomal 12p12 deletion was suspected by bioinformatics analysis using off-target reads (arrow). Array comparative genomic hybridization confirmed 12p.12.2p12.1 deletion. Optic nerve atrophy was thought to be related to a whole deletion of *SOX5* gene. Because a whole deletion of the *ABCC9* gene located nearby to the *SOX5* gene was detected, monitoring of regular cardiac function was recommended.

**Figure 6 genes-13-00027-f006:**
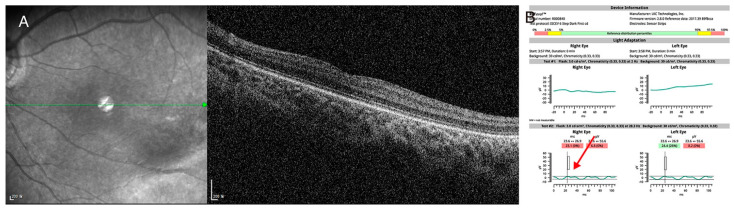
(P6) A case of dual diagnosis identified by targeted next-generation sequencing (**A**) Spectralis optical coherence tomography showed mild temporal retinal dragging with shallow fovea pit. (**B**) Hand-held electroretinogram showed severely attenuated light-adapted 3.0 response and reduced double peak in light-adapted 30 Hz flicker response (red arrow), which was consistent with incomplete congenital stationary night blindness. Targeted next-generation sequencing showed the hemizygous c.1910+1G>A canonical splice site variant in *CACNA1F* gene and c.3833G>A:p.(Trp1278*) nonsense variant in *LRP5* gene. Dual diagnosis of familial exudative vitreoretinopathy and incomplete congenital stationary night blindness was made in this patient, and regular monitoring of bone densitometry was recommended.

**Table 1 genes-13-00027-t001:** Disease-associated variants identified in 86 patients with definite diagnosis.

No.	Initial Clinical Diagnosis	Gene	Mutations	Zygosity	Segregation Analysis	gnomAD (MAF)	*CADD*	Previous Literature (PMID)	ACMG Classification	Accession ID for Transcript
1	Cone-rod dystrophy	*ABCA4*	c.1958G>A:p.(Arg653His) c.3470T>G:p.(Leu1157*)	Compound heterozygous	NA	10/280000 None	25.9 43	10711710 29975949	LP P	NM_000350.2
2 ^d^	LCA	*AHI1*	c.2174G>A:p.(Trp725*)	Homozygous	NA	4/248916	43	25445212	P	NM_001134831.1
3 ^d^	Laurence-Moon syndrome	*BBS1*	c.908delT:p.(Val303Glyfs*29) c.1285C>T:p.(Arg429*)	Compound heterozygous	Paternal Maternal	None 3/251446	34 40	32165824 ^e^ 12677556	P P	NM_024649.4
4 ^d^	IIN	*CACNA1F*	Exon 13-23 deletion	Hemizygous	NA	None	NA	34064005 ^e^	P	NM_005183.2
5 ^d^	LCA	*CACNA1F*	c.2175_2179delins CATCATGTATGATGGTATCATGGCATT:p.(Gly726Ilefs*61)	Hemizygous	Maternal	None	28.3	34064005 ^e^	P	NM_005183.2
6 ^d^	IIN	*CACNA1F*	c.1910+1G>A	Hemizygous	NA	None	21	34064005 ^e^	P	NM_005183.2
		*LPR5*	c.3833G>A:p.(Trp1278*)	Heterozygous	NA	None	45	Novel	LP	NM_002335.2
7 ^d^	IIN	*CACNA1F*	c.1301C>T:p.(Ala434Val)	Hemizygous	Maternal	None	17.47	28002560	LP	NM_005183.2
8 ^d^	IIN	*CACNA1F*	c.1910+1G>A	Hemizygous	NA	None	21	34064005 ^e^	P	NM_005183.2
9 ^d^	LCA	*CEP290*	c.1666delA:p.(Ile556Phefs*17) c.3904C>T:p.(Gln1302*)	Compound heterozygous	Maternal Paternal	245/174532 None	31 37	16909394 25445212	P P	NM_025114.3
10	Achromatopsia	*CNGA3*	c.1190G>T:p.(Glu397Val) ^a^ c.1279C>T:p.(Arg427Cys) ^a^ c.553C>G:p.(Leu185Val)	Compound heterozygous	Paternal Paternal Maternal	None 110/281878 1/251394	25.5 33 20.5	18636117 11536077 31144483	LP LP LP	NM_001298.2
11 ^d^	Stickler syndrome	*COL2A1*	c.3165+1G>A	Heterozygous	NA	None	25.5	34680973 ^e^	P	NM_001844.4
12 ^d^	Pierre-Robin sequence	*COL2A1*	c.2680-3C>G	Heterozygous	NA	None	23.7	34680973 ^e^	P	NM_001844.4
13 ^d^	LCA	*CRX*	c.101-1G>A ^b^ c.122G>A:p.(Arg41Gln) ^b^	Compound heterozygous	Trans by IGV	None 1/31396	32 25.1	32165824 ^e^ 9427255	P P	NM_000554.4
14	Congenital cataract	*CRYGC*	c.173T>C:p.(Leu58Pro)	Heterozygous	NA	None	26	Novel	LP	NM_020989.3
15	RP	*EYS*	c.8805C>G:p.(Tyr2935*) Exon 42-43 duplication	Compound heterozygous	NA	None None	35 NA	22363543 Novel	LP US	NM_001142800.1
16	IIN	*FRMD7*	c.368C>A:p.(Ser123Tyr)	Hemizygous	NA	None	23.8	Novel	LP	NM_194277.2
17	IIN	*FRMD7*	c.575A>C:p.(His192Pro)	Hemizygous	NA	25/182271	25.6	30025138	LP	NM_194277.2
18	IIN	*FRMD7*	c.637G>A:p.(Val213Met)	Heterozygous	NA	None	32	Novel	LP	NM_194277.2
19	IIN	*FRMD7*	c.685C>T:p.(Arg229Cys)	Hemizygous	NA	None	34	17768376	P	NM_194277.2
20	IIN	*FRMD7*	c.772A>G:p.(Lys241Arg)	Hemizygous	NA	None	27.4	31106028	LP	NM_194277.2
21	IIN	*FRMD7*	c.875T>C:p.(Leu292Pro)	Heterozygous	NA	4/183318	27.5	25678693	P	NM_194277.2
22	IIN	*FRMD7*	c.875T>C:p.(Leu292Pro)	Hemizygous	NA	4/183318	27.5	25678693	P	NM_194277.2
23	IIN	*FRMD7*	c.875T>C:p.(Leu292Pro)	Heterozygous	NA	4/183318	27.5	25678693	P	NM_194277.2
24	IIN	*FRMD7*	c.875T>C:p.(Leu292Pro)	Heterozygous	NA	4/183318	27.5	25678693	P	NM_194277.2
25	IIN	*FRMD7*	c.875T>C:p.(Leu292Pro)	Hemizygous	NA	4/183318	27.5	25678693	P	NM_194277.2
26	IIN	*FRMD7*	c.875T>C:p.(Leu292Pro)	Hemizygous	NA	4/183318	27.5	25678693	P	NM_194277.2
27	IIN	*FRMD7*	c.875T>C:p.(Leu292Pro)	Hemizygous	NA	4/183318	27.5	25678693	P	NM_194277.2
28	IIN	*FRMD7*	c.875T>C:p.(Leu292Pro)	Hemizygous	NA	4/183318	27.5	25678693	P	NM_194277.2
29	IIN	*FRMD7*	c.875T>C:p.(Leu292Pro)	Hemizygous	NA	4/183318	27.5	25678693	P	NM_194277.2
30	IIN	*FRMD7*	c.875T>C:p.(Leu292Pro)	Hemizygous	NA	4/183318	27.5	25678693	P	NM_194277.2
31	IIN	*FRMD7*	c.875T>C:p.(Leu292Pro)	Hemizygous	NA	4/183318	27.5	25678693	P	NM_194277.2
32	IIN	*FRMD7*	c.875T>C:p.(Leu292Pro)	Hemizygous	NA	4/183318	27.5	25678693	P	NM_194277.2
33	IIN	*FRMD7*	c.886G>T:p.(Gly296Cys)	Hemizygous	NA	None	34	30015830	LP	NM_194277.2
34	IIN	*FRMD7*	c.901T>C:p.(Tyr301His)	Hemizygous	NA	None	26.7	29145603 ^e^	LP	NM_194277.2
35	IIN	*FRMD7*	c.1016C>G:p.(Ser339Cys)	Hemizygous	NA	None	34	Novel	LP	NM_194277.2
36	IIN	*FRMD7*	c.1023_1030AGACCTCC: p.(Asp342Leufs*2)	Hemizygous	NA	None	35	Novel	P	NM_194277.2
37	IIN	*FRMD7*	Exon 5 deletion	Hemizygous	NA	None	NA	Novel	P	NM_194277.2
38	Congenital cataract	*FTL*	c.-168G>T	Heterozygous	Maternal	None	20.8	9414300	P	NM_000146.3
39	FEVR	*FZD4*	c.752C>G:p.(Pro251Arg)	Heterozygous	Maternal	None	25.1	Novel	LP	NM_012193.3
40	FEVR	*FZD4*	c.205C>T:p.(His69Tyr)	Heterozygous	NA	138/277634	24.2	15370539	P	NM_012193.3
41	FEVR	*FZD4*	c.470T>C:p.(Met157Thr)	Heterozygous	NA	None	21.8	21097938	P	NM_012193.3
42	Congenital cataract	*GJA3*	c.290T>G:p.(Leu97Arg)	Heterozygous	NA	None	29	Novel	US	NM_021954.3
		*OPA1*	c.449-2A>C	Heterozygous	NA	None	23.6	Novel	P	NM_130832.2
43	Ocular albinism	*GPR143*	c.248T>C:p.(Leu83Pro)	Hemizygous	NA	None	25.9	31106028	LP	NM_000273.2
44	Ocular albinism	*GPR143*	c.360+2T>C	Hemizygous	NA	None	23	Novel	P	NM_000273.2
45	Ocular albinism	*GPR143*	c.925delG:p.(Ala309Profs*24)	Hemizygous	NA	None	34	Novel	P	NM_000273.2
46	Ocular albinism	*GPR143*	c.518C>G:p.(Ala173Asp)	Hemizygous	NA	None	24.2	Novel	LP	NM_000273.2
47	Ocular albinism	*GPR143*	Xp22.3 deletion	Hemizygous	NA	None	-	Novel	P	-
48	Ocular albinism	*GPR143*	c.223_228dupGCTGCC: p.(Ala75_Ala76dup)	Hemizygous	NA	None	8.758	28339057	LP	NM_000273.2
49 ^d^	LCA	*GUCY2D*	c.1991A>C:p.(His664Pro) c.2984G>A:p.(Arg995Gln)	Compound heterozygous	Maternal Paternal	None 1/244998	27 35	28966547 32165824 ^e^	P LP	NM_000180.3
50 ^d^	LCA	*GUCY2D*	c.2649del:p.(Phe883Leufs*13)	Homozygous	NA	None	33	28966547	P	NM_000180.3
51 ^d^	LCA	*GUCY2D*	c.1790G>A:p.(Gly597Glu) exon 4-5 duplication	Compound heterozygous	Paternal De novo	None None	27.4 -	29068479 32165824 ^e^	LP LP	NM_000180.3
52 ^d^	IIN	*GUCY2D*	c.1978C>T:p.(Arg660*) c.2947C>A:p.(Pro983Thr) ^a^ c.2960G>C:p.(Gly987Ala) ^a^	Compound heterozygous	Paternal Maternal	1/251328 None None	40 25.3 28.4	10766140 32165824 ^e^ 32165824 ^e^	P LP LP	NM_000180.3
53	CCDD	*KIF21A*	c.1067T>C:p.(Met356Thr)	Heterozygous	Maternal	None	25.4	14595441	LP	NM_001173464.1
54	FEVR	*LRP5*	c.607G>A:p.(Asp203Asn)	Heterozygous	Paternal	None	32	16252235	P	NM_002335.2
55	Congenital cataract	*NHS*	c.1117C>T:p.(Arg373*)	Heterozygous	NA	None	38	14564667	P	NM_198270.2
56 ^d^	LCA	*NMNAT1*	c.275G>A:p.(Trp92*) c.709C>T:p.(Arg237Cys)	Compound heterozygous	Paternal Maternal	None 14/282780	38 35	32165824 ^e^ 22842227	LP P	NM_022787.3
57 ^d^	LCA	*NMNAT1*	c.196C>T:p.(Arg66Trp) c.709C>T:p.(Arg237Cys)	Compound heterozygous	Paternal Maternal	22/282832 14/282780	35 35	22842227 22842227	P P	NM_022787.3
58 ^d^	Optic atrophy	*NR2F1*	c.91_93dupCGC:p.(Arg31dup)	Heterozygous	NA	None	19.92	34466801 ^e^	LP	NM_005654.4
59 ^d^	Optic atrophy	*NR2F1*	c.513C>G:p.(Tyr171*)	Heterozygous	NA	None	37	34466801 ^e^	LP	NM_005654.4
60 ^d^	IIN	*NYX*	c.182_183insT: p.(Cys62Valfs*53)	Hemizygous	Maternal	None	32	34064005 ^e^	P	NM_022567.2
61	Unexplained visual loss	*NYX*	c.38-1_38delGCinsTT: p.(Ala13Vafs*102)	Hemizygous	Maternal	None	14.49	ClinVar	P	NM_022567.2
62	Optic atrophy	*OPA1*	c.1240A>C:p.(Thr414Pro)	Heterozygous	NA	None	26.5	26905822	LP	NM_015560.2
63	Optic atrophy	*OPA1*	c.795_798delTGAC: p.(Asp266Cysfs*41)	Heterozygous	NA	None	35	Novel	P	NM_015560.2
64	*PAX6* phenotype	*PAX6*	c.383G>A:p.(Arg128His)	Heterozygous	NA	None	34	30167917	LP	NM_000280.4
65	*PAX6* phenotype	*PAX6*	c.607C>T:p.(Arg203*)	Heterozygous	NA	None	36	7550230	P	NM_000280.4
66	*PAX6* phenotype	*PAX6*	c.397G>T:p.(Glu133*)	Heterozygous	NA	None	38	16712695	P	NM_000280.4
67	*PAX6* phenotype	*PAX6*	c.702T>A:p.(Tyr234*)	Heterozygous	NA	None	36	Novel	P	NM_000280.4
68	*PAX6* phenotype	*PAX6*	c.362C>T:p.(Ser121Leu)	Heterozygous	NA	None	25.9	23734086	LP	NM_000280.4
69	*PAX6* phenotype	*PAX6*	c.607C>T:p.(Arg203*)	Heterozygous	NA	None	37	7550230	P	NM_000280.4
70	*PAX6* phenotype	*PAX6*	c.702T>A:p.(Tyr234*)	Heterozygous	NA	None	36	Novel	P	NM_000280.4
71	Achromatopsia	*PDE6C*	c.1771G>A:p.(Glu591Lys) c.2269C>T:p.(Gln757*)	Compound heterozygous	Paternal Maternal	2/251120 None	33 49	26992781 Novel	P P	NM_006204.3
72	Achromatopsia	*PDE6C*	c.85C>T:p.(Arg29Trp) c.712C>T:p.(Arg238*)	Compound heterozygous	Maternal Paternal	6/282886 2/251376	29.3 38	19615668 27124789	P P	NM_006204.3
73	Optic atrophy	*PDHA1*	c.232G>A:p.(Ala78Thr)	Hemizygous	Maternal	None	24.8	Novel	LP	NM_000284.3
74	RP	*PRPH2*	c.708C>G:p.(Tyr236*)	Heterozygous	Paternal	None	37	22863181	P	NM_000322.4
75	CSNB	*RHO*	c.302G>A:p.(Gly101Glu)	Heterozygous	NA	2/250994	25.5	26161267	LP	NM_000539.3
76	Cone dystrophy	*RP1*	c.4196delG:p.(Cys1399Leufs*5) c.6181delA:p.(Ile2061Serfs*12)	Compound heterozygous	NA	None None	23.7 23	25097241 29425069	P P	NM_006269.1
77	LCA	*RPGRIP1*	c.3565_3571delCGAAGGC: p.(Arg1189Glyfs*7)	Homozygous	NA	4/249114	35	18682808	P	NM_020366.3
78 ^d^	*PAX6* phenotype	*SLC38A8*	c.692G>A:p.(Cys231Tyr) c.964C>T:p.(Gln322*)	Compound heterozygous	Paternal Maternal	None 2/248656	27.2 37	32744312 ^e^ 32744312 ^e^	LP P	NM_001080442.1
79 ^d^	Ocular albinism	*SLC38A8*	c.995dupG:p.(Trp333Metfs*35) c.1214+5G>C	Compound heterozygous	Paternal Maternal	None None	22.9 23.2	32744312 ^e^ 32744312 ^e^	P LP	NM_001080442.1
80 ^d^	*PAX6* phenotype	*SLC38A8*	c.558C>A:p.(Tyr186*) c.1078_1104del: p.(Ala360_leu368del)	Compound heterozygous	Maternal Paternal	None None	58 16.29	32744312 ^e^ 32744312 ^e^	P LP	NM_001080442.1
81	Oculocutaneous albinism	*SLC45A2*	c.220T>C:p.(Trp74Arg)	Heterozygous	Maternal	None	26.7	Novel	US	NM_016180.3
82 ^d^	LCA	*SPATA7*	c.388C>T:p.(Gln130*) c.1160+1G>A	Compound heterozygous	Maternal Paternal	2/249908 None	35 33	32165824 ^e^ 32165824 ^e^	P P	NM_018418.4
83 ^d^	LCA	*TUBB3*	c.967A>G:p.(Met323Val)	Heterozygous	De novo	None	25.3	33921132 ^e^	LP	NM_006086.4
84	Oculocutaneous albinism	*TYR*	c.929dupC:p.(Arg311Lysfs*7) c.1037-7T>A	Compound heterozygous	NA	11/251178 242/280983	22.9 18.81	2511845 8217557	P P	NM_000372.4
85	Usher syndrome	*USH2A*	c.2802T>G:p.(Cys934Trp) c.11389+3A>T	Compound heterozygous	Son ^c^	57/282482 2/250762	26.6 22.7	21686329 28714225	P LP	NM_206933.2
86	Usher syndrome	*USH2A*	c.8559-2A>G	Homozygous	NA	8/251134	34	32093671	P	NM_206933.2
87	X-linked retinoschisis	*VPS13B*	c.6200T>A:p.(Leu2067*) c.9530_9531del: p.(Ala3177Valfs*18)	Compound heterozygous	NA	3/250746 None	39 27.5	15141358 Novel	P P	NM_017890.4
88	Corneal dystrophy	*ZEB1*	c.2034_2035delAA: p.(Pro680Phefs*5)	Heterozygous	NA	None	23.8	Novel	P	NM_030751.5
89	Corneal dystrophy	*ZEB1*	c.1576delG:p.(Val526*)	Heterozygous	NA	None	23.9	Novel	P	NM_030751.5
90	Optic atrophy	*SOX5*	12p12.2p12.1 deletion	Heterozygous	De novo	None	NA	Novel	P	NM_006940.4

*ACMG*: American College of Medical Genetics, *CADD*: combined annotation-dependent depletion, *CCDD*: congenital cranial dys-innervational disorder, *FEVR*: familial exudative vitreoretinopathy, *IGV*: integrative genomic viewer, *IIN*: idiopathic infantile nystagmus, *gnomAD*: genome aggregation dataset, *LCA*: Leber congenital amaurosis, *LP*: likely pathogenic, *MAF*: minor allele frequency, *NA*: not available, *P*: pathogenic, *RP*: retinitis pigmentosa, *US*: uncertain significance. ^a^ These two variants were existed in cis, confirmed by manual inspection through integrative genomic viewer. ^b^ These two variants were existed in trans, confirmed by manual inspection through integrative genomic viewer. ^c^ Segregation was confirmed by proband’s children. ^d^ Previously reported patients by the authors. ^e^ Novel, but previously reported by the authors.

**Table 2 genes-13-00027-t002:** Eight patients who were received precision care after targeted next-generation sequencing.

Patient No.	Sex/Age	Clinical Diagnosis before NGS Testing	Diagnosis after NGS Testing	Genes (Mode of Inheritance)	Nucleotide Changes	Amino Acid Changes	Management	Accession ID for Transcript
6	M/4.3	IIN	CSNB	*CACNA1*^1^ (*XL*)	c.1910+1G>A	-	Dual diagnosis	NM_005183.2
			FEVR	*LRP5*^1^ (*AD*)	c.3833G>A	p.(Trp1278*)	Bone densitometry monitoring	NM_002335.2
11	M/0.8	Stickler syndrome	Stickler syndrome	*COL2A1*^1^ (*AD*)	c.3165+1G>A	-	Prophylactic cryotherapy to prevent retinal detachment	NM_001844.4
12	F/10.9	Pierre-Robin sequence	Stickler syndrome	*COL2A1*^1^ (*AD*)	c.2680-3C>G	-	Prophylactic cryotherapy to prevent retinal detachment	NM_001844.4
38	M/7.8	Congenital cataract	Hyperferritinemia-cataract syndrome	*FTL*^1^ (*AD*)	c.-168G>T	-	Avoid unnecessary phlebotomy or medical investigation	NM_000146.3
47	M/5.1	Ocular albinism	Xp22.33p22.2 deletion	*GPR143*^1^ *CLCN4*, *KAL1* *NLGN4X*, *STS* (*XL*)	-	-	Hypotropic hypogonadism investigation	NM_000273.2
54	F/4.8	FEVR	FEVR	*LRP5*^1^ (*AD*)	c.607G>A	p.(Asp203Asn)	Bone densitometry monitoring	NM_002335.2
73	M/3.8	Optic atrophy	Pyruvate dehydrogenase E1-α deficiency	*PDHA1*^1^ (*XL*)	c.232G>A	p.(Ala78Thr)	Ketogenic diet Thiamine treatment	NM_000322.4
90	F/8.1	Optic atrophy	12p12.2p.12.1 Xp22.2p22.13 deletion	*SOX5*^1^ (*AD*) *ABCC9* (*AD*)	-	-	Regular monitoring of cardiac function	NM_006940.4 NM_005691.3

AD, autosomal dominant; CSNB, congenital stationary night blindness; F, female; FEVR, familial exudative vitreoretinopathy; IIN, idiopathic infantile nystagmus; M, male; XL, X-linked. ^1^ Genes were associated with ophthalmological phenotypes.

**Table 3 genes-13-00027-t003:** Medically or surgically actionable genes in inherited eye diseases.

Phenotype	Phenotypic MIM	MIM Gene	Genes	Inheritance	Ophthalmic Phenotype	Medical or Surgical Action
Abetalipoproteinemia	200100	157147	*MTTP*	AR	Pigmentary retinal degeneration	Vitamin A and E supplement
Ataxia with isolated vitamin E deficiency	277460	600415	*TTPA*	AR	Pigmentary retinal degeneration	Treatment with vitamin E
Blepharophimosis-Ptosis-Epichantus Inversus syndrome	110100	605597	*FOXL2*	AD	Blepharophimosis, ptosis, epicanthus inversus	Refer to endocrinologist for premature ovarian failure
Cerebrotendinous xanthomatosis	213700	606530	*CYP27A1*	AR	Congenital cataract	Chenodeoxycholic acid and statins
Congenital Cataracts	613763	123590	*CRYAB*	AD	Congenital cataract	Dilated cardiomyopathy screening
Congenital Cataracts	607330	601637	*CYP51A1*		Congenital cataract	Check sterol profiling
Episodic ataxia type 2	108500	601011	*CACNA1A*	AD	Episodic nystagmus	Acetazolamide for ameliorating nystagmus
Familial exudative vitreoretinopathy	133780	603506	*LRP5*	AD	Temporal retinal dragging	Refer to endocrinologist to monitor bone mineral density
Galactokinase deficiency	230200	604313	*GALK1*	AR	Congenital cataract	Restriction of lactose and galactose intake
Hyperferritinemia-cataract syndrome	600886	134790	*FTL*	AD	Congenital cataract	Avoid unnecessary repeated phlebotomy
Knobloch syndrome	267750	120328	*COL18A1*	AD	High myopia	Brain MRI to detect occipital encephalocele
Lathosterolosis	NA	602286	*SCD5*	AR	Congenital cataract	Cholesterol reducing agent Ultrasound monitoring of the liver Liver transplant may be required
Leber congenital amaurosis	204100	180069	*RPE65*	AR	Nystagmus Retinal degeneration	Gene therapy (voretigene neparvovec-rzyl)
Pyruvate dehydrogenase E1-α deficiency	312170	300502	*PDHA1*	XL	Optic atrophy Strabismus	Ketogenic diet Thiamine treatment
Refsum Disease	266500	602026 601757	*PHYH*, *PEX7*	AR	Pigmentary retinal degeneration	Diet free of phytol, phytanic acid, or their precursor, or plasmapheresis
Retinoblastoma	180200	614041	*RB1*	AD	Intraocular tumor	Serial detail examination of fundus
Stickler syndrome type I	108300	120140	*COL2A1*	AD	High myopia Vitreoretinal degeneration	Prophylactic cryotherapy to prevent retinal detachment
Stomatin-deficient Cryohydrocytosis	608885	138140	*SLC2A1*	AD	Congenital cataract	Ketogenic diet

AD, autosomal dominant; AR, autosomal recessive; MIM, mendelian inheritance in man; NA, not available; XL, X-linked.

## Data Availability

Not applicable.

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
