# Peer review of "Precision Medicine through Next-Generation Sequencing in Inherited Eye Diseases in a Korean Cohort"

_genes, 2021, doi:10.3390/genes13010027_

Round 1
Reviewer 1 Report
The author described a retrospective study where 149 patients with inherited eye diseases were screened for mutation in known disease genes using a capture panel of 429 genes. Molecular lesion has been found for 60% of the patients with 5.4% patient benefits from the diagnosis in terms of care. In addition, revision of the initial clinical diagnosis was made for 15% of the patients. Overall the study is designed with a translational angle and has demonstrated the utility of molecular diagnosis in caring patients with inherited eye diseases. The analysis is thorough, and the data is convincing with experimental validations. My comments are below
- It is interesting that the solving rate is higher in patients with family history than in singleton cases. I am wondering what is the author’s explanation for this observation. Along the line, is the solving rate for patients with systemic phenotype higher or lower than patients with non-syndromic phenotype?
- Since many of the putative pathogenic mutations identified are novel, it would be important to perform segregation test. It is not clear if segregation tests have been conducted for which families. Genotype of individual in addition to the proband in the pedigree should be labelled.
- For the noncanonical splicing site mutations, please check SplicAI score
- In supplement table S5, for reported mutant, please add the reference pubmed ID.
- The molecular diagnostic rate differs significantly depending on the disease type. Therefore, it would be information to provide statistics on the solving rate for each disease type.
Author Response
#Reviewer 1
The author described a retrospective study where 149 patients with inherited eye diseases were screened for mutation in known disease genes using a capture panel of 429 genes. Molecular lesion has been found for 60% of the patients with 5.4% patient benefits from the diagnosis in terms of care. In addition, revision of the initial clinical diagnosis was made for 15% of the patients. Overall the study is designed with a translational angle and has demonstrated the utility of molecular diagnosis in caring patients with inherited eye diseases. The analysis is thorough, and the data is convincing with experimental validations. My comments are below
R1-1. It is interesting that the solving rate is higher in patients with family history than in singleton cases. I am wondering what is the author’s explanation for this observation. Along the line, is the solving rate for patients with systemic phenotype higher or lower than patients with non-syndromic phenotype?
Author’s response: Thank you for your comment. Our study showed that molecular diagnostic rate is higher in patients with family history than in singleton cases. This could be probably due to de novo variants locating in non-coding regions. De novo mutation rate for SNVs in human 1.0 to 1.8 x 10-8 per nucleotide per generation with substantial variation among families. Because coding region accounts for only 1-2% of genomic regions, de novo variations usually located in intron or regulatory regions. Therefore, de novo variations, which cause aberrant splicing or have reduced expression of certain gene located in deep intron or regulatory regions, might not be detected in targeted panel sequencing.
In addition, solving rate for patients with systemic phenotype was 17 out of 23 (73.9%). Solving rate for patients with non-syndromic phenotype was 73 out of 126 (57.9%). So, solving rate for patients with systemic phenotype is higher than patients with non-syndromic phenotype.
R1-2. Since many of the putative pathogenic mutations identified are novel, it would be important to perform segregation test. It is not clear if segregation tests have been conducted for which families. Genotype of individual in addition to the proband in the pedigree should be labelled.
Author’s response:
We presented the results of segregation analysis in table 2 (formerly table S5) and figure S1.
R1-3. For the noncanonical splicing site mutations, please check SpliceAI score.
Author’s response: As you recommended, SpliceAI scores was presented in the text of figure S3. All delta scores were 0.90 or higher.
R1-4. In supplement table S5, for reported mutant, please add the reference pubmed ID (PMID).
Author’s response: We add the PMID in previously reported variants in table 2 (formerly table S5).
R1-5. The molecular diagnostic rate differs significantly depending on the disease type. Therefore, it would be information to provide statistics on the solving rate for each disease type.
Author’s response: Thank you for the comment. We added this in the result section and figure 1.
Reviewer 2 Report
The manuscript submitted by Moon et al. describes a retrospective analysis of genetic results obtained from a cohort of 149 Korean patients with inherited eye diseases. In particular, emphasis is put on genetic findings that allow for precision medicine (actionable genes).
The results of this targeted study are worth communicating to the scientific community and agree with other targeted studies on inherited eye disease, providing a complete molecular diagnosis for approx. 60% of the investigated cohort.
The manuscript is generally well written and structured. However, some suggestions are made below that might improve the quality and scientific depth of this manuscript.
General remarks:
- Title: Optional: Addition of “in a Korean cohort” at the end might be useful, as identified variants can be population-specific or enriched in certain populations.
- Abstract: the percentage of males is less relevant in the abstract. I would rather emphasize the variety of phenotypes.
- Table 1: can you add the transcript used for the c.nomenclature of variants, similar to Table S5?
- Although the emphasis of this manuscript is to highlight findings that allow for precision care/medicine, the general molecular findings in this manuscript are also worth communicating in greater detail. The first and second paragraphs of the results section and/or the discussion sections could be enhanced by the following details:
- Provide a bit more information on the specific subtypes (and %) of inherited eye disease (IED) included in this study. This could also be added graphically to figure 1, or in a separate figure (similarly to figure S3, but with subdivision of syndromic and non-syndromic IRD). In addition, it would be interesting to see the percentage solved per phenotypic category, which could be added within the same figure (see point below). The top mutated genes in the studied cohort can be briefly mentioned in the text.
- It would be interesting to have a more in-depth results/discussion on:
- The solved cohort: freq. solved based on initial phenotype (see point above)? Emphasis was already put on the comparison between syndromic vs. non-syndromic cases and simplex/familial cases, but less on the differences in solved percentage between the larger phenotypic categories.
- The unsolved patients: How many are mono-allelic cases with a class 4/5 in an autosomal recessive gene? It would be worth highlighting that a second, undetected, (novel) non-coding variant /SV can be expected in those cases. The number of identified novel non-coding variant in inherited eye disease is rapidly increasing (esp. for certain genes such as ABCA4).
- Table 2: As this is not a finding of this article, I would move this to supplements and rather add a table/figure with the general (summarized) genetic findings of the studied cohort.
- Figure S4: Could you adapt this, so that the figure is fully visible?
- Line 193/194: If both SOX5 and ABCC9 are encompassed in the same deletion, the sentence should be worded a bit differently to make this clear.
Textual remarks:
Below I have added just a few textual changes that can be made, but the manuscript would benefit from having a general check on textual errors.
Line 29: multiple locus variants was found > were found
Line 61/62: unclear > “when clinical phenotype matches a single genetic cause”
Line 133/134: unclear > “are listed in online supplemental tables”
Line 146: “we identified a total of “
Line 149: “medically actionable”
Line 169: “a whole gene deletion of “
Line 193 & 194/ add “a” in front of deletion.
Line 198: remove “was found”
Line 201: “a/the hemizygous variant”
Line 252: “which reported that multiple molecular diagnoses could be made in 4.9% of cases in exome sequencing”
Line 329: “Our approach helps to avoid” rather than “We thought that this approach may help”.
Author Response
#Reviewer 2
The manuscript submitted by Moon et al. describes a retrospective analysis of genetic results obtained from a cohort of 149 Korean patients with inherited eye diseases. In particular, emphasis is put on genetic findings that allow for precision medicine (actionable genes).
The results of this targeted study are worth communicating to the scientific community and agree with other targeted studies on inherited eye disease, providing a complete molecular diagnosis for approx. 60% of the investigated cohort.
The manuscript is generally well written and structured. However, some suggestions are made below that might improve the quality and scientific depth of this manuscript.
General remarks:
R2-1. Title: Optional: Addition of “in a Korean cohort” at the end might be useful, as identified variants can be population-specific or enriched in certain populations.
Author’s response: Thank you for your comment. We added “in a Korean cohort” at the end of the title.
R2-2. Abstract: the percentage of males is less relevant in the abstract. I would rather emphasize the variety of phenotypes.
Author’s response: We revised the sentence to “Among 149 patients, 38 (25.5%) had a family history, and this cohort includes heterogeneous phenotype including anterior segment dysgenesis, congenital cataract, infantile nystagmus syndrome, optic atrophy, and retinal dystrophy.”
R2-3. Table 1: can you add the transcript used for the c.nomenclature of variants, similar to Table S5?
Author’s response: We add the transcript used for the c.nomenclature of variants on the table 1.
R2-4. Although the emphasis of this manuscript is to highlight findings that allow for precision care/medicine, the general molecular findings in this manuscript are also worth communicating in greater detail. The first and second paragraphs of the results section and/or the discussion sections could be enhanced by the following details:
Provide a bit more information on the specific subtypes (and %) of inherited eye disease (IED) included in this study. This could also be added graphically to figure 1, or in a separate figure (similarly to figure S3, but with subdivision of syndromic and non-syndromic IRD). In addition, it would be interesting to see the percentage solved per phenotypic category, which could be added within the same figure (see point below). The top mutated genes in the studied cohort can be briefly mentioned in the text.
Author’s response: Thank you for the comment. We added specific information on disease category of inherited eye diseases in the results section and figure 1. We deleted figure S3. We added the top mutated gene in our corhot. “FRMD7, PAX6, GPR143 variants appeared to be frequently mutated in our cohort.”
R2-5. It would be interesting to have a more in-depth results/discussion on:
The solved cohort: freq. solved based on initial phenotype (see point above)? Emphasis was already put on the comparison between syndromic vs. non-syndromic cases and simplex/familial cases, but less on the differences in solved percentage between the larger phenotypic categories.
The unsolved patients: How many are mono-allelic cases with a class 4/5 in an autosomal recessive gene? It would be worth highlighting that a second, undetected, (novel) non-coding variant /SV can be expected in those cases. The number of identified novel non-coding variant in inherited eye disease is rapidly increasing (esp. for certain genes such as ABCA4).
Author’s response: We added the solving rate based on different disease category in the results and figure 1. In our cohort, only 1 patient had mono-allelic cases with a class 4/5 in an autosomal recessive gene. In one patient (P96) with oculocutaneous albinism, only one heterozygous c.1025C>T:p.(Ala342Val) variant in SLC24A5 gene was identified. Therefore, genome sequencing with functional validation is required to identify hidden genetic variation in this patient. We added this in the discussion section. The other patient had c.2147A>C:p.(His716Pro) in the CASK gene. The two affected males in this family had infantile-onset nystagmus with X-linked recessive inheritance. However, the phenotype is not fully compatible with CASK-induced mental retardation, with or without nystagmus because two affected males had normal intelligence. We classified this patient as unsolved. We discussed this in the manuscript.
R2-6. Table 2: As this is not a finding of this article, I would move this to supplements and rather add a table/figure with the general (summarized) genetic findings of the studied cohort.
Author’s response: Thank you for the comment. Although this table is not a finding of this article, the table give a valuable information to the readers. Therefore, we changed table S5 (main genetic findings) to table 2 and changed table 2 to table 3.
R2-7. Figure S4: Could you adapt this, so that the figure is fully visible?
Author’s response: We modified it to landscape format in the section, so Figure S4 can be fully visible.
R2-8. Line 193/194: If both SOX5 and ABCC9 are encompassed in the same deletion, the sentence should be worded a bit differently to make this clear.
Author’s response: To make the sentence clear, we revised the sentence 'Regular cardiac function is recommended because nearby ABCC9 gene deletion was found by array comparative genomic hybridization.' to 'Because a whole deletion of ABCC9 gene which located in nearby SOX5 gene was detected, monitoring of regular cardiac function is recommended.'
Textual remarks:
R2-9. Below I have added just a few textual changes that can be made, but the manuscript would benefit from having a general check on textual errors.
Author’s response: Thank you for your kindness. We have a general check on textual errors and edited it accordingly.
R2-10. Line 29: multiple locus variants was found > were found
Author’s response: We modified it.
R2-11. Line 61/62: unclear > “when clinical phenotype matches a single genetic cause”
Author’s response: We revised the sentence.
R2-12. Line 133/134: unclear > “are listed in online supplemental tables”
Author’s response: We modified it.
R2-13. Line 146: “we identified a total of “
Author’s response: We modified it.
R2-14. Line 149: “medically actionable”
Author’s response: We modified it.
R2-15. Line 169: “a whole gene deletion of “
Author’s response: We modified it.
R2-16. Line 193 & 194/ add “a” in front of deletion.
Author’s response: We modified it.
R2-17. Line 198: remove “was found”
Author’s response: We modified it.
R2-18. Line 201: “a/the hemizygous variant”
Author’s response: We modified it.
R2-19. Line 252: “which reported that multiple molecular diagnoses could be made in 4.9% of cases in exome sequencing”
Author’s response: We modified it.
R2-20. Line 329: “Our approach helps to avoid” rather than “We thought that this approach may help”.
Author’s response: We revied the sentence.
Round 2
Reviewer 1 Report
The authors have addressed the critics raised by the reviewers. The manuscript is suitable for publication